# Distress and Resilience in the Days of COVID-19: Comparing Two Ethnicities

**DOI:** 10.3390/ijerph17113956

**Published:** 2020-06-03

**Authors:** Shaul Kimhi, Yohanan Eshel, Hadas Marciano, Bruria Adini

**Affiliations:** 1Stress and Resilience Research Center, Psychology Department, Tel-Hai College, Tel Hai 12210, Israel; yochi_eshel@hotmail.com (Y.E.); hmarcia1@univ.haifa.ac.il (H.M.); 2Ergonomics and Human Factors Unit, University of Haifa, Haifa 3498838, Israel; 3Department of Emergency Management and Disaster Medicine, School of Public Health, Sackler Faculty of Medicine, Tel Aviv University, Tel Aviv 6997801, Israel; adini@netvision.net.il

**Keywords:** COVID-19, sense of danger, distress symptoms, individual, community and national resilience, minority and majority groups

## Abstract

The COVID-19 pandemic is an ongoing epidemic of coronavirus disease, caused by severe acute respiratory syndrome, which has spread recently worldwide. Efforts to prevent the virus from spreading include travel restrictions, lockdowns as well as national or regional quarantines throughout the international community. The major negative psychological outcome of the COVID-19 pandemic is the anxiety caused by it. The aim of the present study was to examine the level of concern and the contributions of modes of resilience, well-being and demographic attributes towards decreasing or enhancing anxiety and depression among two samples: Israeli Jews (majority group) and Israeli Arabs (minority group). These random samples included 605 Jews and 156 Arabs who participated in an internet survey. A previous study, which has been conducted in the context of terror attacks, has shown that compared to Israeli Jews, Israeli Arabs expressed a higher level of fear of war and lower levels of resilience supporting personality attributes. The results of the current study indicated a similar pattern that emerged in the context of the COVID-19 pandemic: the Israeli Arabs reported a higher level of distress and a lower level of resilience and well-being.

## 1. Introduction

The Coronavirus disease (COVID-19) which has erupted in China in 2019 is an infectious disease caused by a newly discovered strain of Coronavirus. This pandemic is rapidly spreading worldwide, with constantly growing numbers of morbidity and mortality [1,2]. It has led to severe global disruptions, such as closing schools and universities, partial or total closure on the population enforced by governments, reduced travel and ensuing unemployment and economic difficulties, world-wide stock markets decline, and panic buying due to widespread fears of supply shortages [3]. The COVID-19 illness is very risky for older people who have additional health problems, but statistics show that the percentage of infected people in the population is relatively small, in countries that maintain proper health systems [4]. The vast majority of the general public is not expected to become sick and suffer directly from the COVID-19 symptoms. However, people are liable to be concerned by this pandemic and suffer from it psychologically. They are likely to experience fear, anxiety, and uncertainty, worrying whether they or their dear ones may be susceptible to the virus, become sick and risk death. Furthermore, many individuals may wonder whether they are already affected by this virus although they do not show the typical symptoms caused by it [5]. Although the limitations of social contact which have been introduced and enforced in various countries may slow this pandemic from spreading, they are likely to intensify people’s concerns and apprehension concerning the dangers of this pandemic.

For the vast majority of people who do not regard themselves as sick yet, the COVID-19 epidemic may constitute a major psychological issue which is expressed by increased tension, concerns, and anxiety. These psychological effects are strengthened by a lack of an effective vaccine that may prevent the disease’s contagion or medicine which can cure it and are also influenced by the ambiguity concerning how long it will continue to disrupt individual and public life [6]. The ongoing intensive discussion of this pandemic by the mass media as well as politicians, and grim forecasts concerning its potential disastrous future outcomes, further enhance these negative emotions [7]. The impact of the COVID-19 pandemic on the general public will be assessed, therefore, by the individual level of concerns and anxiety which is raised by it among the general public.

The present study investigated psychological and demographic variables that affected the level of concern among individuals who, as far as we know, have not been affected by the COVID-19. These impacts were examined separately for two populations: a sample of the Israeli Jewish majority, and a sample of its Israeli Arab minority.

In characterizing the Israeli Arab culture, it has been claimed that Western cultures encourage individualism and self-actualization through attributes such as goals, dreams, abilities, personality traits, and talents, whereas the Arab society in Israel generally centers on a collectivist culture [8]. This culture enhances the importance of intergroup relationships which help each member to fit in this society. It is based on the belief that all problems and issues should be solved within the extended family, without looking outward for help or advice. Its members are expected to turn to their families for assistance, in any social, economic, or health-related difficulty. Despite changes in the structure of the Arab family due to the modernization process, there has been some continuity in the extended relationships between family members, which has retained the three main family units of this society: the *hamula* (kinship group), the extended family, and the nuclear family. The extended family is still the basic social unit responsible for caring for aged family members and for supporting all its members during distressing times (Haj Yahia-Abu Ahmad, 2006, unpublished doctoral dissertation). We assume that this family structure would also support members of this ethnic group during the current COVID-19 pandemic.

A previous comparison of reactions of Jewish and Arab Israelis to threats of war and acts of terror [9,10] have indicated a similarity between Jews and Arabs in their previous exposure to terror. However, the Arab sample expressed a significantly higher level of fear of war and significantly lower levels of community and national resilience. We assumed, therefore, that Israeli Arabs would also show a higher level of COVID-19 psychological symptoms. Moreover, disastrous events are likely to enhance a continuous sense of danger that strongly and negatively influences the reaction to these adversities [11]. A high sense of danger is positively correlated with distress symptoms [12], and negatively correlated with a sense of coherence [13] and individual resilience [14]. A previous study of responses to war threats had indicated that while a sense of danger increases distress feelings, feeling safe at home decreases anxieties raised by a potential calamity [15]. Accordingly, we assumed that Israeli Arabs would show a higher level of sense of danger and a lower level of feeling safe at home, compared with Israeli Jews. Furthermore, earlier studies indicated that highly threatening and painful events, such as the COVID-19 pandemic, undermine people’s basic sense of security and increase distress symptoms. These symptoms include continuous emotional and behavioral reactions [16] such as depression, anxiety, and grief.

Previous studies have indicated that well-being and individual, community and national resilience contribute to countering distress. Individual resilience theory concentrates on understanding the process through which people overcome traumatic events and calamities experienced by them [17]. Researchers have claimed that resilience combines protective factors that modify, ameliorate, or alter a person’s response to environmental hazards that predispose to a maladaptive outcome [18,19]. For example, a previous study [15,18] has shown that individual resilience positively and significantly predicts better coping with stressful situations. In a different context, individual resilience was found to be negatively associated with fear of war among Israelis [14,15]. Community resilience expresses an identification of individuals with their social system, and their belief that their societal networks will succeed in providing for their needs in times of adversities [19]. Community resilience is defined as the network’s “capability to anticipate risk, limit impact, and bounce back rapidly through survival, adaptability, evolution, and growth in the face of turbulent change” [20] (p. 10). A review of the research presents that community resilience is associated with increased local capacity, social support, and resources, as well as decreased risks, miscommunication, and trauma [21]. National resilience represents the society’s ability to withstand an adversity with its values and institutions remaining intact, as well as the society’s ability to cope with a changing, and sometimes hostile, environment by adapting and readjusting in new and innovative ways [22]. It has been claimed that national resilience is composed of four attributes: patriotism, optimism, social integration, and trust in political and public institutions [23]. National resilience has been positively predicted by a sense of coherence, well-being, and economic conditions [24]. Well-being represents individuals’ perceptions of their quality of life [9].

The present study aimed to examine the level of sense of danger and distress symptoms, the contributions of modes of resilience, well-being and demographic attributes towards decreasing or enhancing anxiety and depression among two samples: Israeli Jews (majority group, N = 605) and Israeli Arabs (minority group, N = 156).

Overall, we expected in the current study to find negative correlations between the distress symptoms and a sense of danger, individual, community, and national resilience, and the well-being, across the two groups. We also expected that Arab Israelis would demonstrate a higher level of distress and a lower level of resilience, compared with Israeli Jewish.

In line with the above discussion, we hypothesized the following: 1. The Arab sample would score higher than the Jewish sample on the level of COVID-19 pandemic distress and the level of sense of danger perceived by its members and would score lower than the Jewish sample on individual and public (community and national) resilience as well as on feeling safe at home. 2. The COVID-19 pandemic concerns expressed by both groups would be positively predicted by their sense of danger, and negatively predicted by their individual resilience. Demographic variables associated with greater life experience would negatively predict this anxiety, whereas a higher level of education, which indicates a better understanding of the complex implications of this disease, will positively predict this anxiety.

## 2. Method

### 2.1. Participants and Procedure

The present study was based on a random internet sample of 605 Jews (299 females) and 156 Arab adults (78 women) who have agreed to participate in the research. They are characterized by a wide range of demographic attributes (see demographic characteristics detailed in Table 1). The data have been collected by the Internet Panel Company that consists of the largest internet panel in Israel (over 100,000 panelists). All data was gathered anonymously, following approval of the IRB of the Tel Aviv University for the reliability and validity of the on-line questionnaire, see [25]. All participants signed informed consent forms before filling out the questionnaires.

The participants were randomly sampled from the large internet panel and accessed the questionnaire that was distributed to their e-mail address. At the beginning of the questionnaire, a brief explanation was presented, informing them that the participation in the study is voluntary and the completion of the questionnaire constitutes an agreement to participate in the study. The explanation included the email address of two researchers, inviting the respondents to approach them should the questionnaire cause them any discomfort. The collected data was uploaded to SPSS (version 23) and analyzed to identify differences between groups based on Matt Whitney U test and multiple regressions, conducted separately for each ethnic group.

### 2.2. Materials

#### 2.2.1. Individual Resilience

Individual resilience was measured by a 10-item Connor-Davidson scale [26] portraying individual feelings of ability and power in the face of difficulties. This scale was rated by a 5-point response scale ranging from 1 = not true at all, to 5 = generally true. Significant correlations were found between this scale and emotional intelligence, life satisfaction, self-esteem, and positive affect; and a negative significant correlation was found with negative affect [27]. Cronbach’s alpha reliabilities of this scale were α = 0.87 and α = 0.91, for the investigated Jewish and Arab samples, respectively.

#### 2.2.2. Community Resilience

Community resilience was assessed by a short version of 10 items of the CCRAM scale [28]. For this study, we changed the scale each time the word ‘security crisis’ emerged for the ‘Coronavirus crisis’. This tool covered five main issues: social trust, social support, leadership, emergency preparedness and attachment to place (e.g., “I trust the decision-makers in my community”). Items of this scale were rated by a 5-point scale ranging from 1 (does not agree at all) to 5 (totally agree). The Cronbach alpha reliabilities of this scale were α = 0.92 and α = 0.91 for the Jewish and Arab samples, respectively.

#### 2.2.3. National Resilience

National resilience was measured by the NR-13 instrument [29] that pertains to trust in national leadership, patriotism, coping with national crises, and belief in social justice. The 13 items have been rated by a scale ranging from 1 = Does not agree at all, to 6 = Very highly agree. Cronbach’s alpha reliability of this resilience in the present study was α = 0.88.

#### 2.2.4. Sense of Danger

A seven-item Sense of Danger Scale, which is based on Solomon and Prager [30] scale, referred to as a lingering sense of danger in the context of security threats, was used to assess the sense of danger. In the current study we modified the threat from security to the COVID-19 pandemic threat (e.g., “To what extent are you worried about the increase of the COVID-19 global crisis?”). Also, we added the item “To what extent are you worried that we will not be able to overcome the COVID-19 crisis before many citizens in our country die from this disease?” Responses were rated on a Likert-like scale ranging from 1 (not at all) to 5 (very much). The Cronbach alpha reliabilities of this scale were α = 0.83 for both Arab Jewish samples.

#### 2.2.5. Distress Symptoms (BSI)

The level of individual distress symptoms, in the context of the COVID-19 pandemic, was determined by nine items about anxiety and depression out of the Brief Symptom Inventory [31]. This inventory was scored by a Likert scale ranging from 1 (not suffering at all) to 5 (suffering very much). (e.g., “How much do you suffer from feelings of a sudden fear with no reason?”). Due to ethical considerations, we did not include the item concerning suicidal thoughts. Cronbach’s alpha for both samples was α = 0.86.

#### 2.2.6. Feeling Safe at Home

This issue was examined by a single item: “To what extent do you feel safe at your home?” The response scale ranged from 1 = Not safe at all, to 5 = Very safe.

#### 2.2.7. Well-Being

Kimhi and Shamai [32] have assessed post-war strength by an individual level of recovery from war. In more peaceful times, this strength has been assessed by ‘My Life Today’ or well-being scale in which people rate their current health, work, social contacts, and their achievements. The sense of coherence, social support, and community resilience has consistently predicted this variable over time [33]. The scale consists of 9 items ranged from 1 = Not good at all, to 6 = Very good. Cronbach’s alpha for both samples was α = 82.

#### 2.2.8. Demographic Variables

Six demographic attributes were examined (Table 1): (a) Age; (b) Gender; (c) Religiosity: This variable was assessed by 1-item with a 4-point scale ranging from 1 = secular to 4 = ultra-orthodox; (d) Family income level: This variable was assessed by 1-item with a 5-point scale ranging from 1=much above-average to 5 = much below average; (e) Educational level: This variable was assessed by 1-item with a 5-point scale ranging from 1 = elementary school to 5 = academic (master’s degree and beyond); (f) Number of children.

## 3. Results

### 3.1. Groups’ Comparisons

Due to the inequality of the participants’ characteristics, Mann-Whitney U tests were used to compare the two participating groups—Jews versus Arabs. The demographic characteristics of the two participating groups (Table 1) indicated that the Israeli Arab participants scored significantly lower than the Israeli Jewish participants on two demographic characteristics: age (*p* < 0.0001) and income (*p* < 0.0001), and scored significantly higher on level of religiosity (*p* < 0.04). Concerning political attitudes, Arabs presented more left-wing political attitudes compared to the Jewish group (*p* < 0.0001). No significant difference was found between these groups according to their educational level.

In agreement with the first hypothesis on the mean differences between the two investigated groups, Mann-Whitney U tests indicated that the Arab respondents expressed a significantly higher level of COVID-19 distress (Jewish, M = 2.34, SD = 0.77, Arab M = 2.83, SD = 0.79, Z = 67.6, z = *p* < 0.001; η^2^ = 0.03). Furthermore, Arab respondents expressed a significantly higher level of sense of danger compared with the Jewish respondents’ (Jewish, M = 2.87, SD = 0.75, Arabs M = 3.54, SD = 0.76, Z = −9.20, *p* < 0.001; η^2^ = 0.11, Figure 1). In contrast, Jewish respondents reported a higher level of well-being (Jewish, M = 4.10, SD = 0.83 Arabs M = 3.39, SD = 0.67, Z = −9.55, *p* < 0.01; η^2^ = 0.12); community resilience (Jewish, M = 3.33, SD = 0.80 Arabs M = 3.10, SD = 0.89, Z = −2.86, *p* < 0.01; η^2^ = 0.01); national resilience (Jewish, M = 3.97, SD = 0.87, Arabs M = 3.58, SD = 1.07, Z = −4.33, *p* < 0.001; η^2^ = 0.02); and feeling safe at home (Jewish, M = 4.16, SD = 4.16, Arabs M = 3.71, SD = 1.02, Z = −5.11, *p* < 0.001; η^2^ = 0.03). Though no significant difference was found between these groups concerning individual resilience, the wellbeing reported by the Jewish respondents was higher than that of the Arab respondents (Jewish, M = 4.11, SD = 0.83, Arabs M = 3.40, SD = 0.67, Z = −9.55, *p* < 0.001; η2 = 0.12). These comparisons are presented in Figure 1.

### 3.2. Multiple Regressions

The second hypothesis on the predictors of COVID-19 distress of the participants was examined by two multiple regression analyses, conducted separately for the Jewish and the Arab samples (Table 2). As expected, a higher sense of danger contributed to increasing the COVID-19 distress in both samples, whereas individual resilience contributed to decreasing this anguish in both groups. Feeling safe at home negatively predicted COVID-19 anxiety in the Jewish sample and had no significant effect on the Arab sample. Two additional variables, age and the number of children, negatively predicted this anxiety in the Jewish group, but not in the Arab group. The educational level of the participants significantly predicted COVID-19 anxiety in both groups. However, higher educational levels predicted a higher level of such anxiety among Jewish participants and negatively predicted it among Arab respondents.

## 4. Discussion

The present study aimed to examine the level of concern and the contributions of modes of resilience, well-being, and demographic attributes towards decreasing or enhancing anxiety and depression in two samples: Israeli Jews and Israeli Arabs. The most prominent result of the current study is the differences we found between the Jewish majority and Arab minority samples. The differences between the two samples refer both to different levels of distress and resilience, and their predictors. Members of the current Arab sample expressed higher levels of distress and perceived danger of the present pandemic. This is interesting to note as theoretically, the COVID-19 pandemic does not pose a greater risk to the minority group compared to the majority population. The COVID-19 endangered most of Israel’s citizens to about the same extent, and the percentage of Jews who were affected by it, so far, was higher than the percentage of sick Arabs. The present results seem to indicate that the higher level of anxiety expressed by the Arab sample, more likely portrays permanent concerns of this minority and perceived inequality, which preceded the COVID-19 [34].

Examining the variables that predicted COVID-19 pandemic anxiety in each of the investigated samples indicated, as expected, that a higher sense of danger, raised by this pandemic, tended to enhance anxiety levels, whereas a stronger individual resilience was inclined to decrease the rate of this apprehension. These effects were replicated in both samples. These results strengthen previous findings that presented a negative relationship between individual resilience and anxiety [35].

The rest of the predictors differentiated between the two groups. Feeling safe at home negatively predicted pandemic anxiety. Arab participants felt less secured at their homes compared to Jewish individuals, and their sense of safety at home was not strong enough to negatively affect their anxiety. These perceptions were most likely derived from the ongoing variability that exists between Israeli Arab and Jewish populations concerning safety perceptions in the different settings, including in their homes as was identified in previous studies [1,17]. These differences were likely to impact their perceived well-being and levels of stress [36]. These results partly correlate with the previous finding of the lower resilience of minorities under stressful conditions [37].

Two additional unlikely characteristics contributed to decreasing the investigated anxiety, but in the Jewish sample only: age of participants and number of children. Regarding the first one, it may be reasoned that in the present context this variable is relevant to the extent to which it represents life experience. More mature individuals had probably more than one opportunity to experience different highly distressing events, which affected their lives and the lives of their children, and an opportunity to learn that in most cases these adversities sooner or later disappear. Furthermore, they are more adept at self-regulating their psychological well-being [3]. These findings are aligned with other reported results indicating that older individuals showed lower levels of concern and anxiety, despite their belonging to a much higher-risk group for contracting COVID-19 [38].

It was expected that the potentially disastrous impacts of the COVID-19 pestilence and the difficulties in finding means for treating it, as well as efficient immunization for it, will be appreciated more readily by individuals with higher education. The present findings indicated that the level of education played the opposite role in predicting this distress in the two investigated groups. More advanced education levels contributed to increased anxiety in the Jewish sample, and at the same time was a significant factor in decreasing this anxiety among Arabs. Wang et al. [3] also found during the current pandemic, that higher education was related to higher levels of depression and anxiety. The assumption was that highly educated professionals are accustomed to a busy lifestyle and the current isolation from routine colleagues, activities, and occupations, contributed to elevated frustrations and anxiety [3,39,40].

The lower national resilience of the Arab participants most probably did not reflect solely the way they perceive the government’s management of the COVID-19 crisis. It is more likely that it represented a permanent struggle concerning their national identity and their identification with the state of Israel. It was argued that Israeli Arabs were expected to determine the extent to which they identified themselves as mainly Palestinians or Israelis, and this awkward choice was likely to lead them to a sense of identity threat [41]. These contrary identities were based on different, and opposing national aspirations, and seemed to disagree on the history of the Israeli-Arab conflict, its root causes, and the role played by each group in initiating and maintaining it [42]. Furthermore, the Israeli Arabs tend to regard themselves as a deprived but non-assimilating minority [43], and we submit that responses of minority groups to all kinds of calamities could serve as indicators for their actual or perceived relative social position.

Beyond these specific group differences, we suggest that different socio-cultural groups are likely to respond differently to ambiguous and complicated situations. The regression analyses indicated which individual characteristic predicted COVID-19 anxiety, and there was no reason to believe that members of either group were aware of the prediction pattern that characterized them. However, it appeared that members of the Jewish community raised inadvertently several potential predictors for their emotional response to this pandemic, whereas members of the Arab public tended to restrict their responses unintentionally and seemed to be satisfied with a smaller number of such predictors. This conclusion was supported by a previous comparison between Israeli Arabs and Jews in predicting individual resilience in light of their experienced security situation at that time [9]. This study indicated that the degree of fear of war, and the level of exposure to terror acts negatively predicted individual resilience in both groups. However, while no additional individual characteristic predicted this resilience in the Arab sample, it was significantly predicted as well by age, level of religiosity, political attitudes, family income, and sense of coherence of the Jewish participants. In both studies, the total impact of the predictors on the predicted variable was quite higher in the Jewish sample compared with the Arab sample.

Our results indicated similarities as well as differences between the characteristics of the present sample and those obtained in previous comparisons conducted between Israeli Jews and Arabs in a security context [1,10]: The Arab respondents of the previous research scored significantly lower than the Jews on individual, community and national resilience scales. Furthermore, the Arab respondent was significantly more religious and held more left-wing political attitudes compared to the Jewish sample. Both samples reported a higher level of sense of danger and distress symptoms, a lower level of individual resilience and higher level of community resilience, compared with the results obtained in previous studies. However, there were no differences between the two studies, regarding national resilience [9].

## 5. Limitations 

The first limitation of this study concerns the internet sample on which this study was based, as we cannot guarantee that it is a representative sample of the Israeli population, even though the sample is large and includes a broad distribution of demographic variables. The second limitation is the fact that our study is a correlational study and does not allow causality inference.

## 6. Conclusions

Our results replicate earlier studies indicating that a minority group is more susceptible to disasters and threats, and less resilient, compared with the majority group. It seems that perceived inequalities that existed before the adversity, increased during an emergency. An additional possible conclusion is that the high degree of uncertainty that characterizes the current COVID-19 crisis greatly increases anxiety and distress symptoms. Based on the current research and further studies we strongly recommend continuous measurement of stress and resilience over time.

## Figures and Tables

**Figure 1 ijerph-17-03956-f001:**
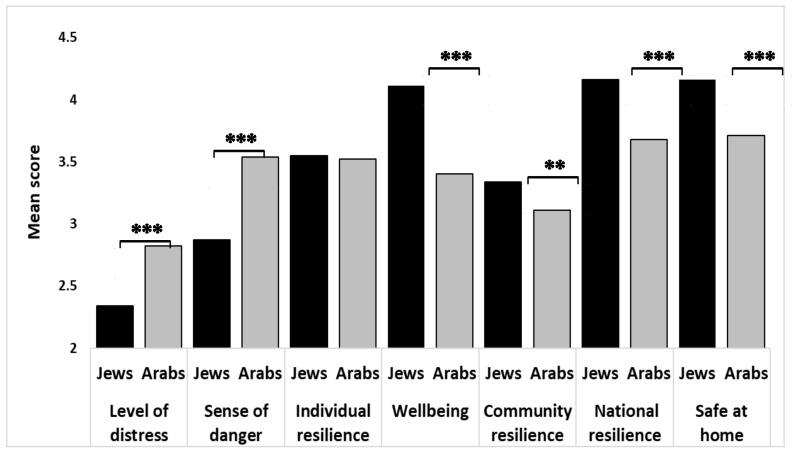
Comparisons of Israeli Jewish and Israeli Arab samples’ mean scores. ** *p* < 0.01, *** *p* < 0.001; Tested with Mann-Whitney U tests.

**Table 1 ijerph-17-03956-t001:** Demographic characteristics and group comparisons comparing mean demographic variables of Jewish and Arab participants.

Variable	Participants	Distribution			*M*	*SD*	*Z*	*η^2^*
(Scale)		Range	n	%		*Z*	*η^2^*	
Age	Jews	18–40	310	51	42.40	15.63	−4.69 ***	0.03
		41–60	193	32				
		61 and above	102	17				
	Arabs	18–40	104	67	35.98	13.55		
		41–60	46	29				
		61 and above	6	5				
Level of Religiosity (scale 1–4)	Jews	Secular	282	47	1.84	0.96	−3.45 ** −3.45 **	
		Traditional	193	32				
		Religious	77	13				
		Orthodox	53	9				0.06
	Arabs	Secular	38	24				
		Traditional	81	52	2.01	0.75		
		Religious	32	20				
		Orthodox	4	3				
Average Family Income (scale 1–5)	Jews	lower	304	51	2.51	1.18	−7.94 ***	
		Average	165	27				
		Higher	109	18				0.06
	Arabs	Lower	126	81	1.69	0.89		
		Average	23	15				
		Higher	7	5				
Political Attitudes (scale 1–5)	Jews	Left	69	11	3.54	0.87	−13.93 ***	
		Center	205	34				
		Right	331	55				0.25
	Arabs	Left	88	56	2.27	0.78		
		Center	65	42				
		Right	3	2				
Educational Level (scale 1–5)	Jews	High School	144	24	3.28	0.98	−0.13	
		Secondary	225	37				
		Academia	236	39				
	Arabs	High School	51	33	3.24	1.03		
		Secondary	28	18				
		Academia	77	49				

** *p* < 0.01, *** *p* < 0.001; Tested with Mann-Whitney U test.

**Table 2 ijerph-17-03956-t002:** Two multiple regression analyses predicting Coronavirus distress of Jews and Arabs, by respondents’ individual and demographic attributes.

Predictors	Jews				Arabs			
	B	Std. Error	Beta	t	B	Std. Error	Beta	t
Sense of Danger	0.34	0.04	0.32	9.11 ***	0.34	0.09	0.32	4.00 ***
Individual Resilience	−0.32	0.04	−0.26	−7.18 ***	−0.23	0.08	−0.25	−3.09 **
Safe at Home	−0.16	0.03	−0.16	−4.47 ***	0.01	0.06	0.01	0.12
Educational Level	0.05	0.03	0.07	1.95	−0.16	0.06	−0.21	−2.64 *
Age	−0.01	0.00	−0.10	−2.50 *	−0.01	0.01	−0.15	−1.37
Number of Children	−0.04	0.02	−0.09	−2.30 *	0.01	0.04	0.04	0.39
R^2^ F			0.32 46.55 ***				0.20 6.17 ***	

* *p* < 0.05, ** *p* < 0.01, *** *p* < 0.001.

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
