# Peer review of "Distress and Resilience in the Days of COVID-19: Comparing Two Ethnicities"

_ijerph, 2020, doi:10.3390/ijerph17113956_

Round 1

Reviewer 1 Report

An interesting and important paper.

I would like to see some reflections on strengths and limitations

F-stats; two quantities that are expected to be roughly equal which the samples are not. How would defense this? Are other stats more relevant?

Author Response

Reviewer 1:

Comment of the reviewer: I would like to see some reflections on strengths and limitations

Response: We have added several lines that address the limitations of the study in the conclusions subsection; please see page 10, lines 385-388 as follows:

  1. Limitations and Conclusions

The first limitation of this study concerns the internet sample on which this study was based, as we cannot guarantee that it is a representative sample of the Israeli population, even though the sample is large and includes a broad distribution of demographic variables. The second limitation is the fact that our study is a correlational study and does not allow causality inference.

Comment of the reviewer: F-stats; two quantities that are expected to be roughly equal which the samples are not. How would defense this? Are other stats more relevant?

Response: According to the reviewer’s comment we have changed F-stats from the previously used analysis of variance to group comparisons using Mann-Whitney-U tests. This was indicated in page 7 lines 249-250 (see the changes at the beginning of the results sections).  Following this, we re-calculated all the findings, as can be found in lines 250 to 273. Overall results indicated the same differences as was previously found when we used the F-stats. In addition, one should note that the Israeli Arabs consist of approximately 20% of the Israeli population and this is why the Arab sample size in the study (156) accounted for approximately 20% of the total sample (761). Accordingly, the difference in the samples’ size is representative of the population. 

Reviewer 2 Report

I am not sure what to do with the results of this article. It is likely true that most people are feeling an increase in stress related to COVID-19, but does it help us to know which groups are most affected if there is no intervention? I also do not think that these results can necessarily be applied to other "minority" groups. For example, Jewish people may be considered a minority in the U.S., but it would be unclear if they would then show less resilience (because they are a minority) or maintain the same level of resilience (because they are Jewish).  

Author Response

Reviewer 2:

Comment of the reviewer: I am not sure what to do with the results of this article. It is likely true that most people are feeling an increase in stress related to COVID-19, but does it help us to know which groups are most affected if there is no intervention? I also do not think that these results can necessarily be applied to other "minority" groups. For example, Jewish people may be considered a minority in the U.S., but it would be unclear if they would then show less resilience (because they are a minority) or maintain the same level of resilience (because they are Jewish). 

Response:  To the best of our knowledge no other study to date examined the level of resilience among minority groups. We believe that this is important information for future preparations to adversities. As explained in this manuscript, the Israeli Arabs are a minority group that is part of the Palestinian people, that are in a state of a prolonged conflict with Israel. Therefore, this minority group should not be compared to other minority groups that have equal rights, such as Jews in the United States. However, there are other minority groups, in different countries in the global community (and even in the USA(, that better fit this pattern, i.e. deprived minorities or minorities that do not identify with the country in which they live. Therefore, from our humble view, the data presented in this manuscript can be generalized and relevant to many other communities or conflicting situations over the globe.   

Reviewer 3 Report

The ms. is a contribution regarding the difference between the Israeli Arab and Jewish,  related to demographic factors and psychological characteristics, such as anxiety, distress and resilience, during  COVID-19 outbreak.

I suggest these changes before the submission:

  • Review the Abstract

Please clarify the aim of the study and report the results clearly.

  • Review the Introduction:
  1. Line 53 please specify the aims of the study, expressly indicating the demographic variables examined. Moreover, please move the aim at the end of the introduction, specifying the sample.
  2. From line 57 to line 123, please modify the text. I suggest removing the subparagraphs, trying to make clear the logical connections that led authors to make certain choices. Now, the reader is not accompanied during the reading. It would probably be appropriate to make reference to the variables, that are under study, by providing information coming from the literature, explaining why they play an important role in this particular period, and so their role on “psychological well-being”. Moreover, it is necessary to clarify what relationships, and what kind of relations (causal?), are present between distress, resilience, risk perception etc.
  3. From line 57 to line 77: this part should be move in the discussion section. In my opinion, to make reference to cultural factors, such as individualism or collectivism, makes no sense, also because it is not a manipulated variable or taken into consideration in the analyses.
  4. Line 81: please remove the sentence “In line with the previous discussion”, you are in the introduction
  • Review the Method section
  1. Line 125: Please change “sample” with “participants”. Moreover, please report in the text for the two groups means and standard deviations about age. Furthermore, use demographic variables subparagraph to describe participants.
  2. Line 132: Please change “instruments” with “materials and procedure”.
  3. Please move any reference to the results, such as correlations or Cronbach alpha reliabilities, from the instrument’s description to result section.
  4. Please use demographic variables as a description of the participants.
  5. Please describe in more details the procedures.
  • Review the Results section
  1. From line 176 to line 183: please move in the description of participants the means reported here, adding standard deviations.
  2. Line 184: Why is the first aim now reported? I should have read it in the introduction
  3. From line 184 to line 193: please report standard deviations
  4. Table 1: descriptive statistics are reported in the text. I suggest removing them from the table. Redundant information.
  5. Line 213: Why is the second aim now reported? I should have read it in the introduction
  • Review the Discussion
  1. Please at the beginning of the introduction make reference to the aims of the study.
  2. I suggest integrating the discussion with a recent paper published on this journal, Germani, A.; Buratta, L.; Delvecchio, E.; Mazzeschi, C. Emerging Adults and COVID-19: The Role of Individualism-Collectivism on Perceived Risks and Psychological Maladjustment. Int. J. Environ. Res. Public Health 2020, 17, 3497.

Author Response

Reviewer 3:

Comment of the reviewer regarding the Abstract: Please clarify the aim of the study and report the results clearly.

Response: We added to the abstract section, the aim of the current study (page 1, lines 16-19) and the main results (page 1, lines 23-25).

Comments of the reviewer regarding the Introduction:

  1. Line 53 please specify the aims of the study, expressly indicating the demographic variables examined. Moreover, please move the aim at the end of the introduction, specifying the sample.

Response: We added the aim of the study, delineated the demographic and psychological variables that were examined, and moved it to the end of the introduction as suggested by the reviewer (page 3, lines 117-120).

  1. From line 57 to line 123, please modify the text. I suggest removing the subparagraphs, trying to make clear the logical connections that led authors to make certain choices. Now, the reader is not accompanied during the reading. It would probably be appropriate to make reference to the variables, that are under study, by providing information coming from the literature, explaining why they play an important role in this particular period, and so their role on “psychological well-being”. Moreover, it is necessary to clarify what relationships, and what kind of relations (causal?), are present between distress, resilience, risk perception etc.

Response: According to the reviewer’s suggestion, we have reedited the introduction section and removed the subparagraphs from line 57 to line 123.  We clarified the relationships between the variables.  See pages 2-3, lines 81 to 115.

  1. From line 57 to line 77: this part should be moved in the discussion section. In my opinion, to make reference to cultural factors, such as individualism or collectivism, makes no sense, also because it is not a manipulated variable or taken into consideration in the analyses.

Response: We think that an article based on a comparison between Jewish and Arab Israelis should be given a brief introduction to Arab society in Israel as part of the introduction to this article and to help the readers understand better the differences between the two groups.

  1. Line 81: please remove the sentence “In line with the previous discussion”, you are in the introduction

Response: We removed this sentence from the manuscript.

Comments of the reviewer regarding the Method section:

  1. Line 125: Please change “sample” with “participants”. Moreover, please report in the text for the two groups means and standard deviations about age. Furthermore, use demographic variables subparagraph to describe participants.

Response:  done; see page 4 line 177 -184; page 6, lines 243-244; page 7, lines 249 – 273

  1. Line 132: Please change “instruments” with “materials and procedure”.

Response: done; see page 4 lines 169 + 186

  1. Please move any reference to the results, such as correlations or Cronbach alpha reliabilities, from the instrument’s description to the result section.

Response: According to APA style the reference regarding materials such as Cronbach alpha reliability is part of the method section.

  1. Please use demographic variables as a description of the participants.

Response: We have added a referral to Table 1 where all demographic descriptions of participants are described in detail. See page 6, lines 243-245

  1. Please describe in more details the procedures

Response: We have added details of the procedures. See page 4 lines 177-184

Comments of the reviewer regarding the results section:

  1. From line 176 to line 183: please move in the description of participants the means reported here, adding standard deviations.

Response: We have added a standard deviation to this section, according to the reviewer’s comment. See page 7, lines 253 - 273

  1. Line 184: Why is the first aim now reported? I should have read it in the introduction

Response: We understand that the reviewer meant the first hypothesis. The first hypothesis appears at the end of the introduction "1. The Arab sample will score higher than the Jewish sample on level of COVID-19 pandemic distress and the level of sense of danger perceived by its members and will score lower than the Jewish sample on individual and public (community and national) resilience as well as on feeling safe at home". Its mention here is a reference to the former appearance in the introduction, for the reader’s convenience.   

  1. From line 184 to line 193: please report standard deviations

Response: We have added standard deviations to this section, according to the reviewer’s comment. See page 7, lines 253-273

  1. Table 1: descriptive statistics are reported in the text. I suggest removing them from the table. Redundant information.

Response: We believe that the descriptive statistics reported in the text helps the reader and it is desirable information, in spite of its redundancy.

  1. 5. Line 213: Why is the second aim now reported? I should have read it in the introduction

Response: Line 213 (now line 290) refers to the second hypothesis and second aim, which was presented before at the end on the introduction, see: “2.  The COVID-19 pandemic concerns expressed by both groups will be positively predicted by their sense of danger, and negatively predicted by individual resilience. Demographic variables associated with greater life experience will negatively predict this anxiety, whereas a higher level of education, which indicates a better understanding of the complex implications of this disease, will positively predict this anxiety.” Once again, the mentioning of this hypothesis here was implemented as a reference to the former appearance in the introduction, for the reader’s convenience.  

Comments of the reviewer regarding the discussion section:

  1. Please at the beginning of the introduction make reference to the aims of the study

Response: We have added a sentence regarding the aims of the study at the beginning of the discussion section. See page 8, lines 305-307

  1. 2. I suggest integrating the discussion with a recent paper published on this journal, Germani, A.; Buratta, L.; Delvecchio, E.; Mazzeschi, C. Emerging Adults and COVID-19: The Role of Individualism-Collectivism on Perceived Risks and Psychological Maladjustment. Int. J. Environ. Res. Public Health 2020, 17, 3497.

Response: We have added a reference to the above-suggested paper in the discussion section. See page 12, reference number 40

Round 2

Reviewer 1 Report

Good paper!

Author Response

Response to reviewer comments

29-5-20

We thank the reviewer for his helpful comments. There were no comments in this round

Reviewer 2 Report

In line 79, I don't think you mean to say that the Arab group would be expected to have more pandemic symptoms. This makes me think they will have more symptoms of COVID-19 (cough, respiratory distress, fever, etc) but I believe you are talking about psychological symptoms/distress. 

In the new section (lines 79-90) you use the word "will" several times when you should use "would." For example, "We assumed the Iraeli Arbabs WILL show..." should be changed to "would show."

I had to read the sentence in lines 91-92 several times to understand what you were saying. Wording is a little awkward

Line 93-94: Avoid using a word in the definition of another term (you use the word "individual" in the definition of "individual resilience." Same suggestion for community and national resilience. Overall, the paragraph from line 91-111 kind of reads like a dictionary/seems like a regurgitation of facts. Perhaps incorporating examples of what you're actually talking about here (illness, stress, etc.) instead of the examples from the literature about things like performance at university would make it fit in more in the article

The paragraph in lines 203-206 is helpful but reads oddly. 

In lines 253-256 I don't think you need to include all the statistical information that you have in parentheses when talking about the differences in demographics. The information is already presented in the table, so it would be better to just say there was a statistically significant difference in age (P value = x) and income (p value=x). 

Define what constitutes left, center, and right political views

Author Response

Response to reviewer comments

28-5-20

Dear editor,

We would like to thank the editor and the reviewers for the opportunity to resubmit our manuscript, “Distress and resilience in days of COVID-19: Comparing two ethnicities”. We thank the reviewers and the editor for their thoughtful and useful feedback. We’ve revised the manuscript accordingly, as follows:

Sincerely,

Professor Shaul Kimhi

Reviewer 2:

Comment of the reviewer: I

Comment of the reviewer: In line 79, I don't think you mean to say that the Arab group would be expected to have more pandemic symptoms. This makes me think they will have more symptoms of COVID-19 (cough, respiratory distress, fever, etc) but I believe you are talking about psychological symptoms/distress. 

Response:  The reviewer is correct; we replaced “pandemic symptoms” by “psychological symptoms”. See page 2, line 79

Comment of the reviewer: In the new section (lines 79-90) you use the word "will" several times when you should use "would." For example, "We assumed the Israeli Arabs WILL show..." should be changed to "would show."

Response:  We revised the paragraph accordingly, replacing “will” with “would”. See lines 73, 79, 85, 160, 161, 163, 165

Comment of the reviewer I had to read the sentence in lines 91-92 several times to understand what you were saying. The wording is a little awkward

Response: We revised the sentence to make it clear. See lines 91-92 that now express the following: “Previous studies have indicated that well-being and individual, community and national resilience contribute to countering distress.”  

Comment of the reviewer: Line 93-94: Avoid using a word in the definition of another term (you use the word "individual" in the definition of "individual resilience." Same suggestion for community and national resilience.

Response: We revised the sentences so that the term that is explained will not be used within the explanation. See line 93 in which “individuals” were replaced by “people”; lines 99-100 that were replaced by “social system, and their belief that their societal networks”; line 101 in which we replaced “community” by “network”.

Comment of the reviewer: Overall, the paragraph from line 91-111 kind of reads like a dictionary/seems like a regurgitation of facts. Perhaps incorporating examples of what you're actually talking about here (illness, stress, etc.) instead of the examples from the literature about things like performance at university would make it fit in more in the article.

Response:  We have revised the paragraph accordingly. See lines 96-97 that were replaced by “individual resilience positively and significantly predicts better coping with stressful situations”

Comment of the reviewer: The paragraph in lines 203-206 is helpful but reads oddly. 

Response:  The paragraph was edited and rephrased as follows: “National resilience was measured by the NR-13 instrument [29] that pertains to trust in national leadership, patriotism, coping with national crises, and belief in social justice.  The 13 items have been rated by a scale ranging from 1 = Does not agree at all, to 6 = Very highly agree.  Cronbach's alpha reliability of this resilience in the present study was α=.88. see lines 203-206

Comment of the reviewer: In lines 253-256 I don't think you need to include all the statistical information that you have in parentheses when talking about the differences in demographics. The information is already presented in the table, so it would be better to just say there was a statistically significant difference in age (P-value = x) and income (p value=x). 

Response: the statistical information was revised in accordance with the suggestion of the reviewer so that the data itself is presented only in Table 1 and the text now presents only the message and the p-value. See lines 253-258

Reviewer 3 Report

The authors answered all the questions arised. I suggest this minor revision before pubblication: 

  1. move the present paragraph  "Overall, we expected in the current study to find negative correlations between the distress symptoms and a sense of danger, individual, community, and national resilience, and the well-being, across the two groups. We also expected that Arab Israelis will demonstrate a higher level of distress and a lower level of resilience, compared with Israeli Jewish" after line 121 of the introduction.

Author Response

Response to reviewer comments

29-5-20

Dear editor,

We would like to thank the editor and the reviewers for the opportunity to resubmit our manuscript, “Distress and resilience in days of COVID-19: Comparing two ethnicities”. We thank the reviewers and the editor for their thoughtful and useful feedback.

We addressed the comments of the reviewers as described below.

Sincerely,

Professor Shaul Kimhi

Reviewer 3:

Comment of the reviewer: The authors answered all the questions I raised.

Response: We thank the reviewer for once more reviewing our updated manuscript.

Comment of the reviewer: I suggest this minor revision before publication: 

move the present paragraph "Overall, we expected in the current study to find negative correlations between the distress symptoms and a sense of danger, individual, community, and national resilience, and the well-being, across the two groups. We also expected that Arab Israelis will demonstrate a higher level of distress and a lower level of resilience, compared with Israeli Jewish" after line 121 of the introduction.

Response: We changed the order of the sentences accordingly. The paragraph has been moved to lines 118-121